# Stepping beyond Counts in Recovery of Total Knee Arthroplasty: A Prospective Study on Passively Collected Gait Metrics

**DOI:** 10.3390/s23125588

**Published:** 2023-06-14

**Authors:** Cam Fary, Jason Cholewa, Scott Abshagen, Dave Van Andel, Anna Ren, Mike B. Anderson, Krishna R. Tripuraneni

**Affiliations:** 1Epworth Foundation, Richmond 3121, Australia; camdon.fary@unimelb.edu.au; 2Department of Orthopaedics, Western Hospital, Melbourne 3011, Australia; 3Zimmer Biomet, Warsaw, IN 46580, USA; jason.cholewa@zimmerbiomet.com (J.C.); scott.abshagen@zimmerbiomet.com (S.A.); dave.vanandel@zimmerbiomet.com (D.V.A.); na.ren@zimmerbiomet.com (A.R.); mike.anderson@zimmerbiomet.com (M.B.A.); 4New Mexico Orthopaedic Associates, Albuquerque, NM 87110, USA

**Keywords:** total knee arthroplasty, TKA patient, gait assessment, patient monitoring, sensors, recovery curve, accelerometers

## Abstract

Advances in algorithms developed from sensor-based technology data allow for the passive collection of qualitative gait metrics beyond step counts. The purpose of this study was to evaluate pre- and post-operative gait quality data to assess recovery following primary total knee arthroplasty. This was a multicenter, prospective cohort study. From 6 weeks pre-operative through to 24 weeks post-operative, 686 patients used a digital care management application to collect gait metrics. Average weekly walking speed, step length, timing asymmetry, and double limb support percentage pre- and post-operative values were compared with a paired-samples *t*-test. Recovery was operationally defined as when the respective weekly average gait metric was no longer statistically different than pre-operative. Walking speed and step length were lowest, and timing asymmetry and double support percentage were greatest at week two post-operative (*p* < 0.0001). Walking speed recovered at 21 weeks (1.00 m/s, *p* = 0.063) and double support percentage recovered at week 24 (32%, *p* = 0.089). Asymmetry percentage was recovered at 13 weeks (14.0%, *p* = 0.23) and was consistently superior to pre-operative values at week 19 (11.1% vs. 12.5%, *p* < 0.001). Step length did not recover during the 24-week period (0.60 m vs. 0.59 m, *p* = 0.004); however, this difference is not likely clinically relevant. The data suggests that gait quality metrics are most negatively affected two weeks post-operatively, recover within the first 24-weeks following TKA, and follow a slower trajectory compared to previously reported step count recoveries. The ability to capture new objective measures of recovery is evident. As more gait quality data is accrued, physicians may be able to use passively collected gait quality data to help direct post-operative recovery using sensor-based care pathways.

## 1. Introduction

The primary objectives of total knee arthroplasty (TKA) are to alleviate pain and restore mobility and physical function. Early mobilization is a primary focus to enhance recovery following surgery [1]. Although the majority of primary TKA patients expect to return to normal physical activity levels following surgery [2], physical activity does not necessarily increase above pre-operative levels in some populations [2,3]. Further, some arthroplasty patients remain less physically active compared to age-matched controls through the first year post-operatively [4].

The validity of self-reported physical activity questionnaires in assessing physical function recovery following lower extremity arthroplasty is controversial [5]. For example, in the general population, 62% of subjects indicated they met minimum physical activity guidelines via a self-assessed questionnaire, but only 9.6% met the requirements measured objectively through physical activity monitoring [6]. Total joint arthroplasty patients have been documented to overestimate self-reported physical activity levels by as much as 50% [7]. Additionally, many patients overestimate their magnitude of physical function on patient-reported outcome measures (PROMs), such as the Lower Extremity Functional Scale (LEFS) or Western Ontario and McMaster Universities Osteoarthritis Index (WOMAC) physical function subscale [8], and incorrectly conflate reductions in pain and exertion with improved physical function [9].

Free-living monitoring of daily step count with commercially available fitness trackers have been studied as an objective measure of physical activity and functional recovery following TKA [10,11], and recent studies show that step count biofeedback may enhance clinical outcomes post-operatively [5,12]. Step counts are reported to recover as fast as 6 weeks following TKA [11]. However, range of motion deficits and gait asymmetries persist for at least 3 months post-operatively [13,14,15]. Therefore, measuring step count alone provides only a partial indication of physical function recovery following TKA.

Gait analysis is an established and reliable tool to assess human ambulation [16], and the parameters produced have been used to quantify abnormal gait in TKA patients [17]. Historically, collecting gait characteristics was performed in a lab, and most reports compared patients to controls at least 6 months post-operatively. However, lack of pre-operative and continuous data does not allow comparison to pre-operative values to assess the trajectory of recovery or determine if the gait abnormalities are a result of the surgery or patterns that were adopted as a result of osteoarthritis progression pre-operatively [18]. Prior research has also reported significant differences in spatiotemporal gait parameters between measures taken in a lab and in the field [19].

Analysis of gait in patients with osteoarthritis via wearable sensors has nearly doubled in the past decade [20]. Most studies have used multiple inertial measurement unit (IMU: accelerometer, gyroscope, and magnetometer combined in a single unit) sensors placed on the trunk and lower limbs (i.e., foot, ankle, shank, or thigh) to assess spatiotemporal parameters of gait [20]. However, multi-sensor protocols require correct IMU placement by the patient, data to be transferred back to the researcher/clinician, and transformation/analysis of the raw data, which creates limitations in the home setting or with large patient cohorts [21]. Single IMU sensors placed on the trunk have been validated [22,23] and studied in patients with knee osteoarthritis [24,25], but also require data transfer and transformation. Beginning in 2017, some smartphones (i.e., iPhone 8, Apple Inc., Cupertino, CA, USA) were released with a +2 g accelerometer capable of measuring accelerations with output data rates between 0.5 Hz to 1 kHz (LIS331DLH, STMicroelectronics, Kokomo, IN, USA), a 3-axis gyroscope capable of measuring angular velocity at up to 2000 deg/s (L3G4200D, STMicroelectronics, Kokomo, IN, USA), and a magnetometer (AKM8975, AKM Semiconductor, Tokyo, Japan). Smartphone-based IMU data has been used to develop predictive models for adverse post-operative events and frailty detection based on gait in cardiac surgery patients [26,27] and assess abnormal gait in osteoarthritis patients [28,29,30]. In 2020, Apple developed gait metric algorithms and made gait metrics available to iPhone owners with iOS 14, and in June 2021, iOS 15 allowed iPhone owners to electronically share their gait metric data with healthcare providers [31]. Passively collected spatiotemporal data has been used to assess the effect of physical activity on walking speed recovery following TKA [32] and to describe the recovery trajectory of select functional metrics (steps per day, stairs climbed, and gait speed) at specific time intervals (one month, three months, and six months) following TKA [33]. However, to our knowledge, continuous monitoring of gait quality recovery during the early post-operative period has not been performed.

The use of commercially available sensor-based technologies allows for gait analysis during activities of daily living [34] and has the potential to allow for clinician monitoring, timely feedback, and the identification of poor recovery prognosis in the early post-operative period. Understanding the trajectory of gait parameter recovery following TKA is important to help surgeons, physical therapists, and patients develop an understanding of recovery rates and potentially direct rehabilitation. Therefore, the purpose of this study was to assess the recovery of walking speed, step length, double limb support, and asymmetry percentage metrics following TKA. We sought to evaluate these data before and after primary TKA.

## 2. Materials and Methods

This was a secondary analysis of a level II study with prospectively collected data from an ethics approved (WCG IRB # 20182013) global, multicenter prospective cohort study. Participants underwent primary TKA between January 2020 and March 2022 (clinicaltrials.gov: NCT# 03737149). Data reviewed from this study included procedure type, sex, and passively collected gait quality metrics across a 30-week time period. The global study has been described in previous reports and consists of three phases including a pilot phase, a randomized controlled trial (RCT) phase, and a longitudinal cohort phase [35,36,37]. These reports demonstrate data from the initial phases of the study. The current analysis utilizes patients from the cohort phase.

To participate in the global study patients needed to be at least 18 years of age, own an iPhone (iPhone 8 or higher, Apple Inc., Cupertino, CA, USA) capable of pairing with the Apple Watch (series 3 or higher, Apple Inc.) and supporting updates, scheduled for a primary total knee arthroplasty (TKA), and capable of walking with minimal assistance (a single walking stick or single crutch) pre-operatively. Exclusion criteria included patients with substance abuse issues, inflammatory arthropathies, participating in any other surgical intervention, physical therapy, or pain management study that would compromise the results of this study, and patients requiring simultaneous or staged bilateral knee arthroplasties less than 90 days apart. All patients gave written consent to participate.

Patients who met the inclusion and exclusion criteria, provided informed consent, and did not already own an Apple Watch were provided with an Apple Watch and the digital care management application (mymobility^®^ Care Management Platform, Zimmer Biomet, Warsaw, IN, USA). Patients consented to receive the watch for the duration of the study and ethics committees reviewed and determined the retention of the watch upon study exit. Pre-operatively, patients were provided with education and exercise content. Post-operatively, patients were provided with an at-home-based therapy program standard to the surgical institution’s standard of care through the app, beginning at discharge through 90 days post-operative. Patient compliance with the therapy program was tracked through the application, which automatically recorded patient activity and the outcome variable gait parameters, including walking speed, step length, asymmetry percentage, and double limb support percentage, directly. Walking speed is an estimation of the patient’s velocity while walking on flat ground, step length is an estimation of the distance between foot strikes, double support time is an estimation of the percentage of a gait cycle when both feet are in contact with the ground, and walking asymmetry is an estimation of the percentage of the time that asymmetric steps occur within a walking bout [38]. Patients were instructed to carry their phones near hip height in a pocket or waist band and wear their watches whenever possible. The app prompts patients to walk, but no specific guidance was provided through the app regarding frequency, intensity, or duration of walking. Data was automatically collected when the patient took 20 steps in a single direction over a flat surface while the smartphone was located near hip height and coupled with the body (i.e., in a rear or side pocket).

Apple has reported good concurrent validity and absolute error during a six-minute walk test (6MWT) on a pressure mat while carrying an iPhone in the side pocket for walking speed (ICC = 0.93, σerror = 0.9 m/s), step length (ICC = 0.85, σerror = 0.05 m), double support percentage (ICC = 0.65, σerror = 2.91%) and asymmetry percentage (positive predictive rate = 83.4%, false negative rate = 9.8%) with the iPhone [38]. Compared to previously validated inertial measurement units (accelerometer, gyroscope, and magnetometer) placed on both feet and the 5th lumbar vertebrae, Werner et al. recently reported clinically acceptable percent errors (9.8 to 14.8%) and good ICCs (0.76 to 0.85) for walking speed and step length during the 6MWT, respectively. Double support percentage was reported to a have higher, but still clinically acceptable error percentage (18.4%) and a moderate ICC (0.58). Test–retest was conducted one week apart and good to excellent ICCs (0.75 to 0.93) were reported for walking speed, step length, and double support percentage [39].

As a secondary analysis, the study used a convenience sample of all patients in the prospective study that met the inclusion and exclusion criteria. All data are reported as means with standard deviations unless otherwise indicated. A minimum of one day per week was required to be collected to be included in the analysis, and data was grouped as weekly averages. Data collected during the first six weeks pre-operative were averaged to represent pre-operative values. The analysis population selection criteria included: all patients eligible and consented, unilateral TKA, key demographic data collected, follow-up data no shorter than 24 weeks post-operative, and at least one of the involved Health Kit data (gait speed, gait asymmetry, and step length) collected. Recovery was operationally defined as when the respective weekly average gait metric was no longer significantly inferior to the respective pre-operative value. Pre- and post-operative gait metrics were compared using paired samples *t*-tests. Statistical analysis was performed using SAS v9.4 (2013, SAS Institute, Inc., Cary, NC, USA) and significance was assessed at *p* < 0.05.

## 3. Results

A Strobe flow diagram of all patients (*N* = 686, Female: *n* = 431 (62.8%), Age: 63.1 + 8.7 years, BMI: 31.5 + 6.4 kg/m^2^) is presented in Figure 1.

### 3.1. Walking Speed

Compared to pre-operative (1.01 + 0.15 m/s), walking speed was lowest at post-operative week two (0.79 + 0.19 m/s, *p* < 0.001). Walking speed was recovered (Figure 2) at week 21 (1.00 + 0.13 m/s, *p* = 0.063), but never exceeded average pre-operative values.

### 3.2. Step Length

Step length was shortest at post-operative week two (0.55 + 0.09 m, *p* < 0.001) compared to pre-operative (0.60 + 0.08 m). Step length approached pre-operative values at week 24 (0.59 + 0.06 m) but remained statistically (*p* = 0.004) lower than pre-operative values (Figure 3). Given the difference in pre- and post-operative step length at 24 weeks is approximately 0.01 m, it is unlikely that this difference is clinically relevant, and that step length returned to pre-operative values.

### 3.3. Asymmetry Percentage

Pre-operative symmetry percentage was 12.5 + 17.6% and reached the highest value at week 2 (52.6 + 33.1%, *p* < 0.001). Asymmetry recovered at week 13 (13.9 + 17.4%, *p* = 0.230) and was significantly (*p* = 0.001) improved compared to the average pre-operative value at week 19 (11.1 + 14.9%, Figure 4).

### 3.4. Double Limb Support Percentage

Compared to pre-operative (31.4 + 1.5%), double support percentage was greatest at week 2 (32.8 + 2.1%, *p* < 0.001) and remained elevated before recovering at week 24 (31.5 + 1.5%, *p* = 0.089, Figure 5).

## 4. Discussion

The most important outcome from this study was the ability to demonstrate recovery curves of gait quality using free-living, passively collected objective measures. Specifically, the data demonstrate that all measures of gait quality were most negatively affected two-weeks post-operative, and although asymmetry percentage recovered at 13 weeks, walking speed, step length, and double limb support percentage required at least 22 weeks to fully recover.

Monitoring mobility in the home setting has historically suffered from several challenges, including accuracy, technology portability, and data accessibility. Gait speed and symmetry have been used to assess dysfunction and treatment progress in the clinical environment [17]; however, these require patients to travel outside the home and necessitate sophisticated laboratory equipment and staff time. Options outside of the lab have been limited to pedometers and accelerometers worn in the home and have required either the patient or health care provider to retrieve the device and transfer and transform the raw data. The use of smart activity trackers available on near-ubiquitously owned smartphone technology has the potential to allow for clinician monitoring and timely feedback without requiring patient travel. 

The recovery curves presented in this study are similar to other recovery patterns following TKA found in literature, which show that the majority of physical function is recovered within the first three months post-operative [40,41,42,43]. Our results also suggest that the time course for full recovery of function measured via gait quality metrics follows a slower trajectory compared to step count. Step count has been reported to recover within six to seven weeks post-operatively following TKA [5,11,44]. In contrast, we found that deficits in gait metrics persisted for up to 24 weeks in TKA patients. These findings agree with two recent meta-analyses that reported that the volume of moderate to vigorous physical activity and high-frequency gait cycles, both of which are related to gait speed in bipedal locomotion, required at least 6–12 months to exceed pre-operative values in TKA patients [4,45].

Based upon the assessment of pre- and early post-operative step counts, Twiggs et al. [11] recommended using the patient’s own pre-operative physical activity level as an appropriate benchmark to reach by six weeks post-operatively. While this offers a patient-specific goal for recovery, it does not account for the effect of pre-operative pain and deformity on gait dysfunction and walking discomfort, nor how these variables may influence a patient’s self-selected intensity and duration or the efficiency of physical activity [45]. For example, advanced OA patients present with lower limb kinematic and kinetic asymmetries during walking that are not corrected by TKA [13,14,15], and gait asymmetries in step length and excursion remain despite pain alleviation with arthroplasty [13,46]. This is possibly due to neuromuscular dysfunctions that have embedded in motor programming over time [47]. Our results suggest that, when using gait quality metrics to monitor recovery, the patient’s individual pre-operative values may be an appropriate indicator of recovery rather than healthy control matched data; however, the 6-week benchmark suggested by Twiggs et al. [11] for step count may be too short of a timeline for gait quality restoration.

Compared to step counts, gait quality metrics may also provide deeper insights related to the recovery of function and ability to accomplish activities of daily living or participate in recreational activities following TKA. With specific regards to daily living, the mean walking speed used by pedestrians at crosswalks is 1.32 m/s and an absolute minimum of 0.49 m/s is necessary to cross a two-lane street with full time allotment [48]. Regarding recovery of function, OA patients take shorter, but quicker steps to obtain a desired gait speed [49], which is indicative of reduced efficiency [50]. Reducing step length is a compensation to lessen medial knee loading and joint contact forces [51] and allows for greater step widths to compensate for instability [52]. Double limb support percentage is also an indicator of instability [53]. A higher percentage in double support indicates that, when a patient supports their weight on the affected limb, the contralateral limb is also relied upon for support [54]. Lastly, chronic asymmetry may alter the mechanical environment of the joint such that the loads and stressors applied during locomotion become damaging [55], which may chronically increase the risk of secondary osteoarthritis in contralateral lower extremity joints [56,57]. Although these lab-based associations are intriguing, whether they also exist in passively collected gait measures requires further investigation. 

### Limitations

Given the industry sponsorship of this study and the use of industry authors, the potential for bias exists, as does a potential conflict of interest. For transparency, five of the seven authors of this manuscript are employed by Zimmer Biomet, Inc. The disclosures can be found following the conclusion, but it is important to note this risk for bias and conflict of interest within the manuscript. However, the topic of industry authorship itself is controversial. For example, Gotzsche et al. noted that ghost writing in industry studies was common and recommended improved transparency [58]. Further, Matheson stressed the importance of transparency and following the ICMJE guidelines for authorship [59]. The authors on this paper all provided significant contributions as noted in the authors contribution below and the authors wish to remain transparent about their participation. Further, the authors have attempted to remove the branding of the sensors and implants from the message of the paper. The study was implant-agnostic and reporting of specific implants was not warranted, thus reducing any bias associated with implant devices. The manufacturers names have been limited to the first mention in the methods and generic terms are used throughout the manuscript. The primary message of this paper is that sensor-based technologies can be used to passively collect objective gait metrics following primary TKA. The establishment of recovery curves with these metrics is the first step in understanding how to use this data in a clinically meaningful manner. As different sensors may use different algorithms, it is necessary to develop recovery curves for every sensor used for clinical purposes. 

Evaluating passively collected data from digital care management systems has certain inherent limitations. This includes patient-specific behaviors associated with carrying their smartphone; for example, if the phone is on an armband, gait quality metrics will not be collected. However, given this is unique to each patient it seems reasonable that these behaviors would be similar both pre- and post-operatively. There may be some effects of the study on patient behavior, but collection of free-living data from a patient’s smartphone likely eliminates the Hawthorne effect, resulting in gait evaluations similar to everyday life, as opposed to those collected in the lab on a treadmill, as seen in traditional gait analyses. This study is also limited by the lack of control associated with both patient comorbidities and post-operative adverse events. Further analysis is needed to determine the effect these have on gait patterns.

## 5. Conclusions

Using commercially available sensor-based technologies, we identified recovery trajectories of gait quality metrics in TKA patients specific to the study devices. We found the recovery was longer in duration and discordant to those previously reported for step counts or self-reported physical activity. We have demonstrated that two-week postoperative gait metrics are the most inferior in the recovery curve, while most of these gait metrics recover at 24 weeks. The results of this study demonstrate the ability to collect gait quality metrics via commercially available sensor-based technologies and provides recovery curves that may be used to assist sensor-based-technology-assisted care management. Further research, with larger samples, are needed to determine if sensor-based care pathways can improve the quality of care. 

## Figures and Tables

**Figure 1 sensors-23-05588-f001:**
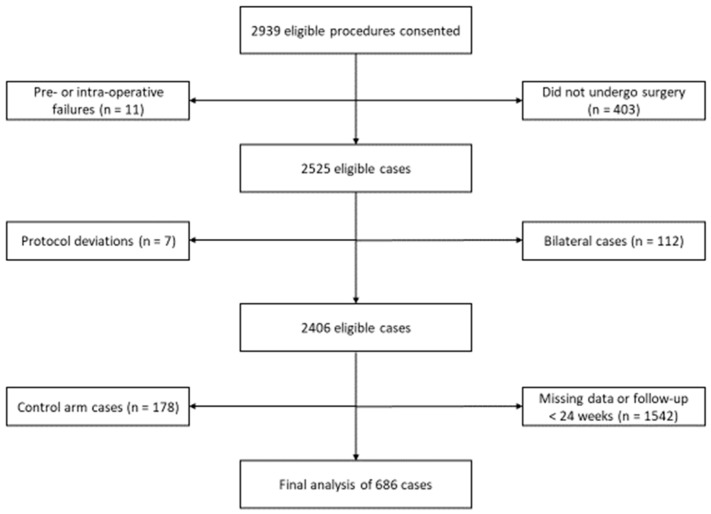
Strobe Flow Diagram.

**Figure 2 sensors-23-05588-f002:**
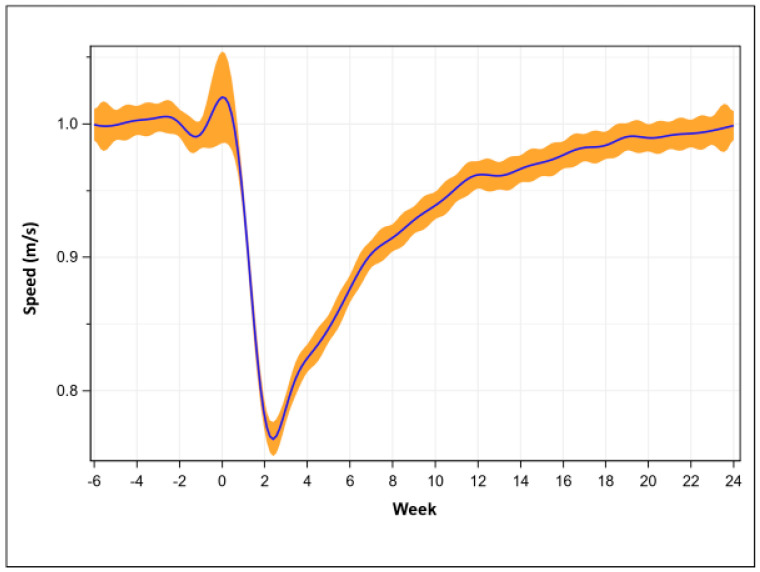
Walking speed recovery trend (mean with 95% confidence intervals).

**Figure 3 sensors-23-05588-f003:**
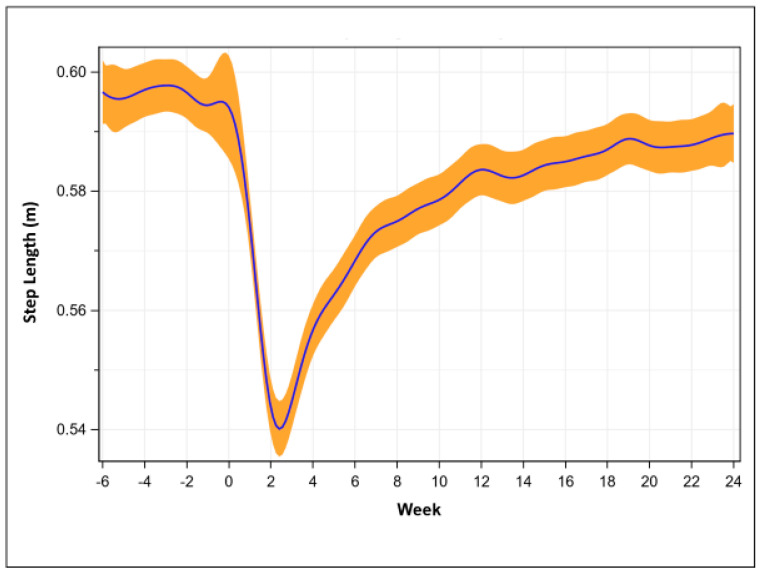
Step length recovery trend (mean with 95% confidence intervals).

**Figure 4 sensors-23-05588-f004:**
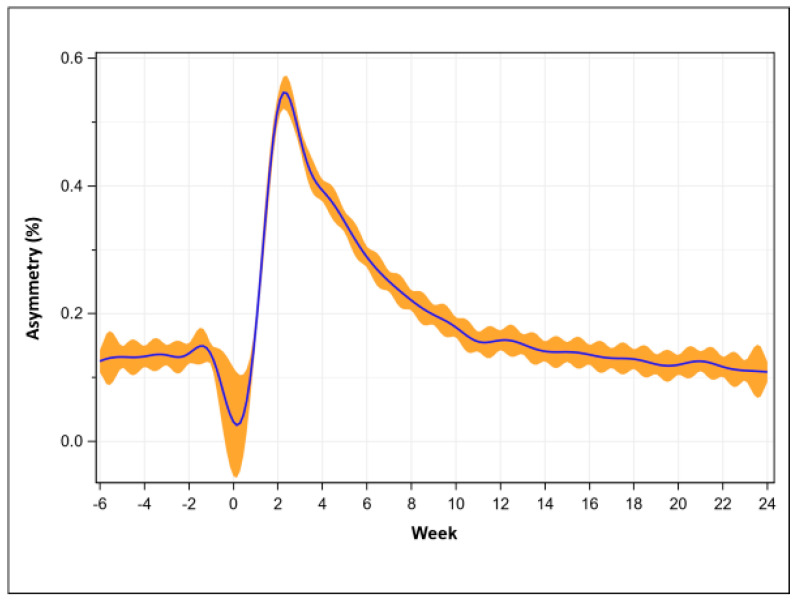
Asymmetry percentage recovery trend (mean with 95% confidence intervals).

**Figure 5 sensors-23-05588-f005:**
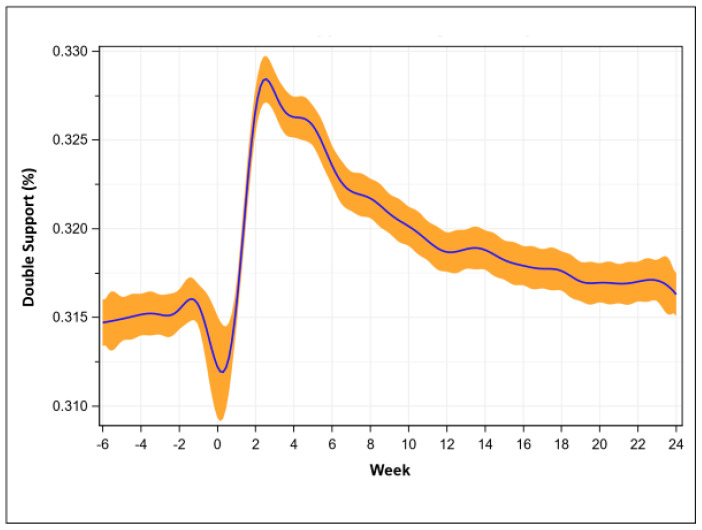
Double limb support percentage recovery trend (mean with 95% confidence intervals).

## Data Availability

The data is unavailable.

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
