# Peer review of "Stepping beyond Counts in Recovery of Total Knee Arthroplasty: A Prospective Study on Passively Collected Gait Metrics"

_sensors, 2023, doi:10.3390/s23125588_

Round 1

Reviewer 1 Report

work of considerable scientific interest given the purpose of the research. the study of the post-surgical gait becomes fundamental for: the improvement of the patient's clinical picture, for the optimization of the rehabilitation method and for the duration of the prosthesis. the possibility of autonomously evaluating the gait with a smartphone allows the reduction of the congestion of surgeries as it will be possible to make continuous evaluations to have a suitable number of data which would improve the final result.

excellent description of materials and methods, good recruitment of patients and the study of these carried out for data collection. of considerable interest is the management software that is simple to apply and standardized for everyone. the programming of the movement gestures required of the patient and standardized is good. the limitations highlighted by the authors who propose further studies to improve the study are also interesting and sincere. The result obtained on the gait negatively influenced after two weeks is interesting, the data on the percentage of asymmetry at 13 weeks are indicative, and the data on walking speed, stride length and the percentage of double limb support which required at least 22 weeks to the complete recovery.

I have no comments on the English language used for this work.

Author Response

Thank you for your positive feedback.

Reviewer 2 Report

Review: Stepping beyond counts in recovery of total knee arthroplasty: a prospective, feasibility study on passively collected gait metrics.

This is a relevant study with potential. Monitoring gait metrics beyond the number of steps in daily life in an important research topic. The authors rightfully pointed out that what happens in a lab-setting using advanced biomechanical research tools not always translates to the clinic and real life. However, I am unconvinced that this manuscript should be published in its current form. In my opinion, this manuscript is too vague to judge the validity in using the presented metrics. Moreover, it is not specified how the metrics are calculated and with which device (i.e., Apple watch or iPhone). It misses some important aspects in looking at the feasibility of using gait metrics in TKA recovery especially using an iPhone and Apple watch. And in my opinion, the manuscript reads like the references were found to fit a certain narrative, and therefore only provides a narrow view of the current literature and limitations.  

Below, I have added some general and specific comments that revolve around the abovementioned points that should be addressed.

General comments

I have some major concerns and questions with using an iPhone and Apple watch. First, an iPhone and Apple watch are very expensive. So how would this work in general? Is good recovery and monitoring only for the rich? Can it be extrapolated to other devices? How would this work in a rehabilitation setting? Does this app involve the intervention of physical therapists/surgeons? If so, whose responsibility is it to provide the iPhone and Apple watch?

You state that all participants receive should have an iPhone and receive an Apple watch. However, it is not clear which device captures the metrics. I understood it that the smartphone captures the data, so why would every individual receive an apple watch? And if the Apple watch was provided as incentive to participate than this should be specified in the manuscript. If the watch had specific functions, then this too needs to be clarified.

Inclusion criteria were patients who had surgery with a Zimmer Biomet product. What does this mean? The type of prothesis? Or some operation tools? Plus, why? You are measuring whether it is feasible to passively collect gait metrics, right? You do not pass any judgement on whether recovery is going well and if these Zimmer Biomet devices are superior to other devices. So why exclude a lot of people (i.e., 962 people) who received surgery using other devices? This does not make sense, especially not concerning generalisability of your product and you miss a lot of potential data.

Line 116-118: “Recovery was operationally defined as when the respective weekly average gait metric was no longer significantly inferior to the respective pre-operative value”.

Where there also measures that improved further than “no longer inferior”. I find this a strange method to quantify recovery. I’m confused about this comparison. Simply because people who will undergo a total knee arthroplasty display an aberrant gait pattern pre-surgery. So why would you define recovery as values similar to pre-op values. You shortly highlight this in the discussion. But not in the introduction or method section.

Plus, it has been shown that physical impairment can improve further than pre-op values. So please specify these points further.

Nowhere in the methods section did you specify how the different metrics were measured and what they mean. Just that Apple has reasonable ICC values. I hope this is in your other papers that were cited but for clarity and transparency, this manuscript should contain a (shortish) description on how this is done. And whether these metrics were validated with golden standard devices.

Figures: Please improve the quality of all your figures.

The flow diagram is unreadable.

Please give more information about your figures. The trend figures: What is this, a single person? Average of all people? If so, please also show some figures that contain the variance of the subjects.

In the conflict-of-interest statement only Zimmer Biomet is mentioned and there is no mention of Apple inc., while the website clearly states that this is a collaboration between the two, and thus, in my opinion it should be mentioned how Apple inc. has been involved in data collection, metric calculation etc..

Specific comments

Line 85: please change gender to sex. I assume you did not ask your participants about their gender.

Line 103-105: “Data was automatically collected when the patient took 20 steps in a single direction over a flat surface while the smartphone was located near the center of mass.”

Please explain how it was determined if the phone was located close to the centre of mass? And how do you define close? Pants pockets close enough? Backpack? Please elaborate. Also please elaborate on how accurate those methods are.

Plus, how was it determined that it was a flat surface? Is a forest surface a flat surface? If not, how do you know?

Line 111: “Data was grouped as weekly averages”. How much data was enough? Just one day? Multiple days? Multiple times over multiple days? Please specify.

Line 107: σerror was 0.05m. But the difference pre-TKA and post-TKA in step length metric is 0.01m. So, can you really conclude that step length did not recover? Plus, the total range in the trend figure is less than 0.05m [0.55m-0.60m]. Therefore, I need to challenge if the step length metric could even be used.

Line 157-160: “Gait speed and symmetry have been used to assess dysfunction and treatment progress in the clinical environment [27]; however, these require patients to travel outside the home and necessitate sophisticated laboratory equipment and staff time”. This is not really true and should be phrased more carefully. As you said, with technological advances there are more opportunities, and these do not always require laboratory equipment. Accelerometers have already been used to assess gait speed and asymmetry in numerous patient populations also in pre- and post-TKA. I implore you to look deeper into the literature.

Line 233: “These data may help develop more specific patient rehabilitation programs for future analysis”. While this is a very general statement used to show some clinical relevance. It feels like an empty statement. “More specific” compared to what? Step count? How do these results show individual trends? Do you see that the people who report to feel poorer, also show poorer metrics? How can you claim that this may help to develop rehabilitation trajectories if those questions are not answered?

Author Response

Reviewer 2: We appreciate your thorough review and feel that by addressing your comments and recommendations the manuscript has been improved. We share your concern about bias noted in both this review and the additional review you provided. We kindly request an objective review of the revisions and hope that they are able to minimize the risk of bias that you have suggested. The intent is to demonstrate the ability to use sensor-based devices as objective measures of outcomes that provide more detail on the qait quality as opposed to activity levels alone.

This is a relevant study with potential. Monitoring gait metrics beyond the number of steps in daily life in an important research topic. The authors rightfully pointed out that what happens in a lab-setting using advanced biomechanical research tools not always translates to the clinic and real life. However, I am unconvinced that this manuscript should be published in its current form. In my opinion, this manuscript is too vague to judge the validity in using the presented metrics. Moreover, it is not specified how the metrics are calculated and with which device (i.e., Apple watch or iPhone). It misses some important aspects in looking at the feasibility of using gait metrics in TKA recovery especially using an iPhone and Apple watch. And in my opinion, the manuscript reads like the references were found to fit a certain narrative, and therefore only provides a narrow view of the current literature and limitations.  

Below, I have added some general and specific comments that revolve around the abovementioned points that should be addressed.

General comments

I have some major concerns and questions with using an iPhone and Apple watch. First, an iPhone and Apple watch are very expensive. So how would this work in general? Is good recovery and monitoring only for the rich? Can it be extrapolated to other devices? How would this work in a rehabilitation setting? Does this app involve the intervention of physical therapists/surgeons? If so, whose responsibility is it to provide the iPhone and Apple watch?

You have asked some very important questions which we will address below. As these questions are general in nature and not the focus of the paper we have opted to respond only in this letter.

  1. an iPhone and Apple watch are very expensive. So how would this work in general?
    • The ownership of smart phones has become commonplace with the Pew Research Center (2019 report) reporting that approximately 5 billion people worldwide owned a smartphone in 2018. They further noted that, 81% of US adults owned a smart phone and 76% of citizens in advanced economies owned smart phones. Presently, it is estimated that 6.9 billion people own a smart phone, equating to global ownership over 80%. Given the general population uses smart phones we feel the generalizability of this paper is reasonable.
      • https://www.pewresearch.org/global/2019/02/05/smartphone-ownership-is-growing-rapidly-around-the-world-but-not-always-equally/#:~:text=For%20example%2C%20a%20median%20of,only%2045%25%20in%20emerging%20economies.
      • https://www.statista.com/statistics/330695/number-of-smartphone-users-worldwide/
    • Degroot et al have reported that the low cost accelerometer models range from $56 to $146 USD for basic models. Additionally, the validity of lower cost accelerometers was questioned. The cost of an Apple Watch series 3 is approximately $150
      • Degroote L, Hamerlinck G, Poels K, Maher C, Crombez G, De Bourdeaudhuij I, Vandendriessche A, Curtis RG, DeSmet A. Low-Cost Consumer-Based Trackers to Measure Physical Activity and Sleep Duration Among Adults in Free-Living Conditions: Validation Study. JMIR Mhealth Uhealth. 2020 May 19;8(5):e16674. doi: 10.2196/16674. PMID: 32282332; PMCID: PMC7268004.

  1. Is good recovery and monitoring only for the rich?
    • As noted, the majority of the population already own a smart phone device. For the subset of patients without a smartphone, a compatible iPhone and Apple watch can be purchased new for approximately $140 and $150, respectively. Restored devices can be purchased for even less. Additionally, the watch is not necessary for the calculation of gait metrics in this specific analysis, only the iPhone.
      • https://www.walmart.com/ip/Apple-iPhone-8-64GB-Space-Gray/402474539?wmlspartner=wlpa&selectedSellerId=101330676&&adid=22222222227402474539_101330676_152791653721_18606250974&wl0=&wl1=g&wl2=c&wl3=658959797405&wl4=pla-2020148428443&wl5=200510&wl6=&wl7=&wl8=&wl9=pla&wl10=737403330&wl11=online&wl12=402474539_101330676&veh=sem&gclid=CjwKCAjwvJyjBhApEiwAWz2nLYHV-UHcli9J_H6_d3qmau_wi1tJHGmeUUBgHqp79kw8iZi3sl2yohoCoBAQAvD_BwE&gclsrc=aw.ds

  1. Can it be extrapolated to other devices?
    • The mymobility app, which contains metrics in addition to gait quality, is available on Android or Apple (iOS) devices, however, gait quality metrics are currently available only to Apple users and further study is necessary to validate algorithms on non-iPhone devices. In this study, patients bringing their own Apple device were approached for participation and provided a Series 3 Apple Watch when necessary. In a commercial setting, watch fulfillment is a function of the commercial agreement between the customer and the manufacturer/application developer and is separate from the study contracts.
    • With regards to the question about applicability, the mymobility app has a Web-accessible dashboard that displays recovery metrics and adherence to the care team for each patient enrolled.
    • Finally, other sensor-based devices may provide similar gait metrics and these findings could be applied to those sensors. However, data on the recovery curves of those sensors is needed as studies have questioned the correlation of metrics between devices.

  1. How would this work in a rehabilitation setting?
    • The application has a Web-accessible dashboard that displays recovery gait metrics and adherence to the care team (surgeon, nurse case managers, physical therapists, etc.)
    • The care team can review the dashboard to evaluate the patient’s progress.
  2. Does this app involve the intervention of physical therapists/surgeons? If so, whose responsibility is it to provide the iPhone and Apple watch?
    • See above (item 4)
    • See above (item 3)

You state that all participants should have an iPhone and received an Apple watch. However, it is not clear which device captures the metrics. I understood it that the smartphone captures the data, so why would every individual receive an apple watch? And if the Apple watch was provided as incentive to participate than this should be specified in the manuscript. If the watch had specific functions, then this too needs to be clarified.

  1. The present study was a secondary analysis of data collected from a larger prospective cohort study. The cohort study investigated features associated with the digital care management platform during the episode of care. One of the objectives of the cohort study that pertains to the present investigation was “To provide a real-world data set on the mymobility cohort, including physiologic monitoring data, that enables ongoing exploratory data analysis and [secondary analyses] that inform future product feature development and research.”
  2. Data collected by the watch, but not included in this manuscript, includes heart rate, heart rate variability, steps, flights of stairs climbed, and stand hours. Some of the data related to the smartwatch has already been published: Tripuraneni et al. J Arthroplasty. 2021 Dec;36(12):3888-389    https://pubmed.ncbi.nlm.nih.gov/34462184/
    1. For transparency in the paper and to be consistent with prior publications, the offering of the watch was noted in this manuscript.
  3. The apple watch was provided to participants to capture step counts and other metrics not reported in this secondary analysis. The ability to provide the watch was approved by individual ethics committees.

Inclusion criteria were patients who had surgery with a Zimmer Biomet product. What does this mean? The type of prothesis? Or some operation tools? Plus, why? You are measuring whether it is feasible to passively collect gait metrics, right? You do not pass any judgement on whether recovery is going well and if these Zimmer Biomet devices are superior to other devices. So why exclude a lot of people (i.e., 962 people) who received surgery using other devices? This does not make sense, especially not concerning generalisability of your product and you miss a lot of potential data.

  1. The initial inclusion criteria of the study were limited to Zimmer Biomet implants, prosthesis type. In a subsequent amendment the application was made device agnostic and as such the study allowed additional devices. However, those devices were added near the end of the enrollment period for this secondary analysis and as such many were excluded for not having the minimum 24-week follow-up. We have re-run the analysis with all patients meeting the other inclusion/exclusion criteria. The results have been updated and the paper is now device agnostic. The patient attrition flow chart has been simplified to exclude these cases in the exclusion criteria for <24-week follow-up.
  2. The inclusion of non-Zimmer Biomet implants has increased the sample size by 210 cases to 686 cases, but had negligible effects on the results and recovery curves.
  3. As you note, the study itself is not about the implant and thus there is no intention to report on the superiority of one implant system over another.

 Line 116-118: “Recovery was operationally defined as when the respective weekly average gait metric was no longer significantly inferior to the respective pre-operative value”.

Where there also measures that improved further than “no longer inferior”. I find this a strange method to quantify recovery. I’m confused about this comparison. Simply because people who will undergo a total knee arthroplasty display an aberrant gait pattern pre-surgery. So why would you define recovery as values similar to pre-op values. You shortly highlight this in the discussion. But not in the introduction or method section.

Plus, it has been shown that physical impairment can improve further than pre-op values. So please specify these points further.

  1. We agree that improvement beyond baseline is the ultimate goal of TKA. However, in the immediate post-operative period, TKA resulted in reduced capabilities, as evidenced by the significant reduction in gait metrics at week two in our data and consistent with published reports on the recovery curve[1]. Therefore, we defined recovery from the surgery as a return to pre-operative (baseline) capabilities. Further, prior studies assessing step count recovery curves following TKA have referred to recovery as the time point when post-operative step counts were similar to pre-operative step counts [2-4]. Therefore, we believe there is precedent to operationally define recovery as recovery from the procedure. Given the precedence, we do not feel it necessary to revise our use of the term recovery.

Nowhere in the methods section did you specify how the different metrics were measured and what they mean. Just that Apple has reasonable ICC values. I hope this is in your other papers that were cited but for clarity and transparency, this manuscript should contain a (shortish) description on how this is done. And whether these metrics were validated with golden standard devices.

  1. The biomechanical model and algorithms used to estimate walking speed, step length, double support and asymmetry percentage have not been made public by Apple Inc.
  2. We have included a brief description of each metric at lines 135-139.
  3. A validation study of Apple’s health kit data was recently published, and we have included this information in the methods section at lines 147-155. Wener et al. Sci Rep. 2023 Apr 1;13(1):5350. https://pubmed.ncbi.nlm.nih.gov/37005465/

Figures: Please improve the quality of all your figures.

  1. Figure quality was lost due to being copied and pasted into the journal’s formatted Microsoft Word document. We have submitted a file of high-quality images with the original submission.  We have also resubmitted a file of updated high-quality images, as we recognize the new images in the word file also appear low quality.

The flow diagram is unreadable.

  1. We have inserted another figure, though the quality is only marginally better due to copy and pasting into Microsoft Word. The editorial office was been provided with a file of high-quality images for each figure at submission, and an updated file with the revised manuscript.

Please give more information about your figures. The trend figures: What is this, a single person? Average of all people? If so, please also show some figures that contain the variance of the subjects.

  1. The recovery curve was a plot of the means from each week. We agree visualizing variance in the dataset is important, and have updated the figures to include 95% confidence intervals and have included mean and 95% confidence interval in the caption. 

In the conflict-of-interest statement only Zimmer Biomet is mentioned and there is no mention of Apple inc., while the website clearly states that this is a collaboration between the two, and thus, in my opinion it should be mentioned how Apple inc. has been involved in data collection, metric calculation etc..

  1. Zimmer Biomet is the study sponsor and provided all funding associated with this study, including the provision of Apple Watches. Apple, Inc. did not participate in data collection or analysis. The Apple HealthKit metrics evaluated in this study were collected through subjects’ device permission-sharing to the mymobility application for viewing in their surgeon’s dashboard and aggregation for study purposes. Apple, Inc. provided resources for application developers to import and understand their Mobility Metrics. This information can be found here: https://developer.apple.com/documentation/coremotion
    1. This information has been added to the conflicts of interest section.

Specific comments

Line 85: please change gender to sex. I assume you did not ask your participants about their gender.

  1. Edited 

Line 103-105: “Data was automatically collected when the patient took 20 steps in a single direction over a flat surface while the smartphone was located near the center of mass.”

Please explain how it was determined if the phone was located close to the centre of mass? And how do you define close? Pants pockets close enough? Backpack? Please elaborate. Also please elaborate on how accurate those methods are.

  1. We have updated the manuscript according to the conditions provided by Apple for phone placement requirements (at the hip (hip clip), in a front or back pocket, or in a waist bag) in their validation study and referenced in a recently published validation study.
    1. https://www.apple.com/healthcare/docs/site/Measuring_Walking_Quality_Through_iPhone_Mobility_Metrics.pdf
    2. Wener et al. Sci Rep. 2023 Apr 1;13(1):5350. https://pubmed.ncbi.nlm.nih.gov/37005465/

Plus, how was it determined that it was a flat surface? Is a forest surface a flat surface? If not, how do you know?

  1. The iPhone and health kit are user devices, and Apple Inc. has not published their algorithms related to health kit data, thus we cannot comment on what conditions are required to constitute “flat surface”. However, the algorithms developed by Apple Inc. are based on identifying flat surfaces during the gait cycle.

Line 111: “Data was grouped as weekly averages”. How much data was enough? Just one day? Multiple days? Multiple times over multiple days? Please specify.

  1. One day per week was the minimum requirement, as many patients only had data for one day in the first week post-operative (inserted at lines 159-161). The frequency of number of days contributing to the weekly average across the entire data set for each metric is as follows:

Gait speed

Asymmetry

Double Support

Step Length

Number of Days

%

%

%

%

1

2.4%

7.5%

3.6%

2.4%

2

3.2%

8.1%

3.9%

3.2%

3

3.8%

9.8%

4.5%

3.8%

4

5.6%

12.2%

6.4%

5.6%

5

8.5%

15.5%

9.3%

8.5%

6

17%

19.7%

17.8%

17%

7

59.4%

27.3%

54.5%

59.3%

Line 107: σerror was 0.05m. But the difference pre-TKA and post-TKA in step length metric is 0.01m. So, can you really conclude that step length did not recover? Plus, the total range in the trend figure is less than 0.05m [0.55m-0.60m]. Therefore, I need to challenge if the step length metric could even be used.

  1. This is an excellent point, though it was statistically significant it is not likely to be clinically relevant. As such, we have noted this in the results. Lines 24-25, lines 191-192.

Line 157-160: “Gait speed and symmetry have been used to assess dysfunction and treatment progress in the clinical environment [27]; however, these require patients to travel outside the home and necessitate sophisticated laboratory equipment and staff time”. This is not really true and should be phrased more carefully. As you said, with technological advances there are more opportunities, and these do not always require laboratory equipment. Accelerometers have already been used to assess gait speed and asymmetry in numerous patient populations also in pre- and post-TKA. I implore you to look deeper into the literature.

  1. We appreciate this constructive feedback and hope the following will better explain what we were attempting to communicate here.
    1. We have updated this sentence as follows to better reflect the state of the art: “Gait speed and symmetry have been used to assess dysfunction and treatment progress in the clinical environment [5]; however, these require patients to travel outside the home and necessitate sophisticated laboratory equipment and staff time. Options outside of the lab have been limited to pedometers and accelerometers worn in the home and have required either the patient or health care provider to retrieve the device and transfer and transform the raw data.”
    2. While accelerometers and ICUs have been used to measure gait pre- and post-TKA, we are unaware of any studies that have passively collected gait quality metrics and continuously analysed these metrics over the episode of care.
      1. For example, Aggarwal et al. used an accelerometer to assess physical activity at 48 and 72 months after TKA on a subset of patients (n=32) for seven consecutive days. They used a common accelerometer for research settings and their analysis was limited to assessing daily activity levels.
        1. Agarwal V, Smuck M, Shah NH. Quantifying the relative change in physical activity after Total Knee Arthroplasty using accelerometer based measurements. AMIA Jt Summits Transl Sci Proc. 2017 Jul 26;2017:463-472. PMID: 28815146; PMCID: PMC5543365.
      2. Other studies have either had patients report to a laboratory and used IMUs or have assigned patients perform standardized measurements of gait capacity by providing multimedia instructions for self-administered test protocols.
  • Other studies have not assessed the recovery curve continuously, but instead at pre-determined time intervals, such as pre-operative, one, three and six months post-operatively. Our study fills in gaps by describing how the gait is affected in between these time points.
    1. Kagan, R., et al., The Recovery Curve for the Patient-Reported Outcomes Measurement Information System Patient-Reported Physical Function and Pain Interference Computerized Adaptive Tests After Primary Total Knee Arthroplasty. J Arthroplasty, 2018. 33(8): p. 2471-2474.
  1. Degroot et al also reported that the low cost models range from $56 to $146 USD for basic models. Additionally, the validity of lower cost accelerometers was questioned. They did not report on the cost of the high quality research based accelerometers.
    1. Degroote L, Hamerlinck G, Poels K, Maher C, Crombez G, De Bourdeaudhuij I, Vandendriessche A, Curtis RG, DeSmet A. Low-Cost Consumer-Based Trackers to Measure Physical Activity and Sleep Duration Among Adults in Free-Living Conditions: Validation Study. JMIR Mhealth Uhealth. 2020 May 19;8(5):e16674. doi: 10.2196/16674. PMID: 32282332; PMCID: PMC7268004.

Line 233: “These data may help develop more specific patient rehabilitation programs for future analysis”. While this is a very general statement used to show some clinical relevance. It feels like an empty statement. “More specific” compared to what? Step count? How do these results show individual trends? Do you see that the people who report to feel poorer, also show poorer metrics? How can you claim that this may help to develop rehabilitation trajectories if those questions are not answered?

  1. The primary purpose of this study was to investigate the feasibility of passively collecting objective measures with commercially available smart technologies and to define the recovery trajectory of gait metrics following TKA. We’ve demonstrated that passive collection is feasible and have reported trends found within this cohort related to the recovery curve.  Correlations between PROMs and gait metrics were outside the scope of the present study. Although it would be interesting to comment on the development of rehabilitation more specific than step counts or PROMs (i.e.: KOOS), as there appears to be a disconnect between how a patient objectively functions versus how they “feel” (see the work of Stratford and Kennedy), such a claim would be speculative.  We have therefore changed the sentence to:  
    1. “The results of this study demonstrate the feasibility of not only collecting gait quality metrics via a commercially available smart technologies, but also provides recovery curves that may be used to assist smart technology assisted care management. Further research, with larger numbers, are needed to determine if sensor-based care pathways can improve the quality of care.”

Reviewer 2 Additional Comments:

The inappropriate citation has mainly to do with the amount of citations they used. In the methods section they refer to their papers that explains different sections of the protocol, but they have cited 3 papers, without specifying what they use out of those prior reports. This is incredibly vague and from that statement you can't derive which "sections of the protocol" they refer to.

  1. We agree that the methods were vague on this topic and could be improved. Additionally, we removed the reference on the total hip publication as it is not appropriate in this paper. Further, we have attempted to explain the overall premise of the study regarding these references. Additional details about the cohort study were presented in the subsequent paragraph. The 1st paragraph of the methods has been amended regarding these citations as follows: “The global study has been described in previous reports and consists of three phases including a pilot phase, a randomized controlled trial (RCT) phase, and a longitudinal cohort phase [6, 7]. These reports demonstrate data from the initial phases of the study. The current analysis utilizes patients from the cohort phase.”
    1. This study was performed as a secondary analysis and the referencing of the publications prior is consistent with good publication practices.
      1. DeTora, L.M., et al., Good Publication Practice (GPP) Guidelines for Company-Sponsored Biomedical Research: 2022 Update. Annals of Internal Medicine, 2022. 175(9): p. 1298-1304

The ethical concerns I have are the costs of the equipment they use in combination with the inclusion criteria.

  1. We understand the concern over costs of the equipment, and the ethics surrounding them. However, costs associated with these devices are described above in the reviewers original comments (an iPhone and Apple watch are very expensive. So how would this work in general?). Additionally, those who met criteria and did not have an apple watch consented to receive and use the watch for the duration of the study. All sites received ethics review and approval of these methods and sites were required to follow the ethics guidance on the final distribution of the watch.

Apple products are very expensive, and if this product will be used in clinical settings, this will have consequences for the quality of care. By using Apple, you are limiting the number of people that can benefit from these types of measurements and makes the quality of rehab dependent on the amount of money you have. Nowhere in their report to they come back to this. This, in my opinion, is also part of the feasibility of using these devices and products.

  1. This is an interesting comment, and it is unreasonable to think that the cost of the product would be detrimental to the quality of care. As already discussed in the reviewer’s initial comments, the cost of the sensor-based technologies used in this study is comparable to other sensor-based devices and necessary to have quality metrics. Further, the current standard of care does not include the use of sensor-based technologies and that is what is being tested in this study. Until the standard of care adopts sensor-based technologies the quality of care cannot be impacted by them. Additionally, though we show the sensors can be used to passively capture objective gait metrics, the clinical implication and use of this data to guide care requires further evaluation and development of sensor-based care pathways. This has been added to the conclusion of the document. Additionally, as we move toward more and more technology assisted care these devices must be studied to determine if their use is not only feasible, but beneficial. The ability to passively collect these data with commercially available devices and the ability to identify specific recovery curves associated with them makes this technology capable of being used pending further investigation. This information has been added to the conclusion. Lines 314-315

Secondly, they only include people who have been treated with Zimmer products, the company the authors work for. This is a massive conflict of interest. While they do specify this conflict, nowhere in the actual manuscript text is it referred to why they do this. This should be mentioned in there for transparency.

  1. We respect your concerns over the conflict of interest posed by including industry authors. After thorough discussion we agree that this could result in a conflict and have re-run the analysis with all patients meeting the other inclusion/exclusion criteria. The results have been updated and the paper is now device agnostic.

On company website it states that there is a partnership between Zimmer Biomet and Apple. This is not mentioned in the manuscript. What this partnership involves, how much apple has to gain from a positive report etc. cannot clearly be understood from this manuscript. I advice that this too needs to be specified for transparency, as this too, involves feasibility of using these devises in a clinical setting.

  1. This relationship has been described in the conflicts of Interest section. We do not feel the relationship with Apple, Inc. had any impact on the results of this paper.

References

  1. Anderson, M., et al., THE RECOVERY CURVE FOR PHYSICAL ACTIVITY FOLLOWING PRIMARY KNEE ARTHROPLASTY USING PASSIVELY COLLECTED OBJECTIVE MEASURES WITH A SMARTPHONE-BASED CARE PLATFORM AND SMART WATCH. Orthopaedic Proceedings, 2021. 103-B(SUPP_9): p. 15-15.
  2. BÄ…czkowicz, D., et al., Gait and functional status analysis before and after total knee arthroplasty. Knee, 2018. 25(5): p. 888-896.
  3. Lebleu, J., et al., Predicting physical activity recovery after hip and knee arthroplasty? A longitudinal cohort study. Braz J Phys Ther, 2021. 25(1): p. 30-39.
  4. Twiggs, J., et al., Measurement of physical activity in the pre- and early post-operative period after total knee arthroplasty for Osteoarthritis using a Fitbit Flex device. Med Eng Phys, 2018. 51: p. 31-40.
  5. Constantinou, M., et al., Spatial-temporal gait characteristics in individuals with hip osteoarthritis: a systematic literature review and meta-analysis. J Orthop Sports Phys Ther, 2014. 44(4): p. 291-b7.
  6. Crawford, D.A., et al., 2021 Mark Coventry Award: Use of a smartphone-based care platform after primary partial and total knee arthroplasty: a prospective randomized controlled trial. Bone Joint J, 2021. 103-B(6 Supple A): p. 3-12.
  7. Tripuraneni, K.R., et al., A Smartwatch Paired With A Mobile Application Provides Postoperative Self-Directed Rehabilitation Without Compromising Total Knee Arthroplasty Outcomes: A Randomized Controlled Trial. J Arthroplasty, 2021. 36(12): p. 3888-3893.

Reviewer 3 Report

Thank you for the opportunity to review your manuscript. The use of modern technology will undoubtedly have an important role to play in future rehabilitation programmes for patients undergoing TKA, so the topic of this research is of current interest.

In this study, 476 participants wore and Apple Smart Watch  for 6 months (6 weeks before surgery and 24 weeks afterwards) to track their spatio-temporal parameters during early recovery via an app on an iPhone. The pre-operative metrics were statistically compared to the post-operative metrics. The authors found the variables to be poorer 2-weeks post-operatively, and recover to pre-operative levels within 24 weeks. While this is an interesting and simple study, I have concerns about the accuracy and validity of the data. Please see my detailed comments below:

Conflict of interest: The role of Zimmer Biomet in this project is unclear. The company funded the study and were apparently involved in analysing the data and writing the manuscript. Further information on the involvement is required, and should perhaps be highlighted within the manuscript for transparency.

Introduction: 

General: The background into the use of smartphones for collecting spatio-temporal parameters is lacking. Please expand this section, as this is a key focus of this study.

Line 56: One of the references cited here is irrelevant as it is a study on total hip arthroplasty

Line 58: The reference here is on total hip arthroplasty and not total knee arthroplasty

Line 77: It is unclear how this is a feasibility study. Please elaborate further on this point. It appears that no variables on the feasibility of using the technology or running a larger-scale study in future were measured. Thus, the aims of this study are not clear.

Methods:

General comment: Was a sample size calculation performed for this study? If so, please include this information in the Methods. If not, please explain why.

General comment: One of my main concerns with this research study is that there is very little information on how the devices provided actually calculate the data measured. Much more detail needs to be included here so that we can understand what the authors are measuring and reporting on and whether the way in which the data was collected was reliable. It should also be clarified whether the devices have been validated in people with total knee arthroplasty, or whether all validation studies to date have been on healthy young adults.

General comment: There is almost no information on the instructions provided to patients with regards to taking part in the study. Were all participants instructed to use the devices for a certain period each day? Were they encouraged use them while exercising? What exactly were the participants told to do for the study? Without this information, it is difficult to interpret the data.

Lines 56-87: It is unclear how this study relates to 'different aspects of the study'. Please explain in detail here what is meant by this statement. It is particularly important to know if the participants were asked to undertake other tasks as part of their involvement that may influence the findings of this study e.g. additional tasks related to their rehabilitation or activity. 

Line 69 & 71: The references are irrelevant to the population in question.

Line 99: What education and exercise content were patients provided with? Are these routinely administered to all patients at the study site, or were these specific to the study participants? If specific, please consider in the discussion how this may have affected the outcomes observed.

Lines 100-101: How was compliance reported? Was this automated i.e. through movement of the smartphone/smart watch, or did you rely on self-reported input from the participants? 

Line 104: How do the devices know if they are near the centre of mass? In other words, if the devices are not near the centre of mass, is the inaccurate data not recorded/deleted? What is the likelihood of erroneous data being included in the analyses?

Line 116-118: Why was recovery defined as a return to baseline when these patients were likely to have pathological walking patterns pre-operatively? 

Results:

Figure 1: It is unclear to me why patients who did not have a Zimmer-Biomet implant were excluded from this analysis, as the research question appears to be completely unrelated to the implant used. This raises conflict of interest issues and ethical issues with regards to participant inclusion in the study.  Please clarify further.

Discussion:

General comment: The results need to be discussed in more detail with regards to previously published research.

Line 158: The reference is not relevant to the population in question. 

Conclusion:

General comment:  It is difficult to know the validity of the conclusion, as the methods by which the data were collected are not described clearly enough in the manuscript. 

Author Response

Reviewer 5. We appreciate your view of the potential of modern technology to play a role in recovery, but more importantly the value you place on transparency and the need to report valid data. We hope our responses provide sufficient information to resolve your concerns.  

Thank you for the opportunity to review your manuscript. The use of modern technology will undoubtedly have an important role to play in future rehabilitation programmes for patients undergoing TKA, so the topic of this research is of current interest.

In this study, 476 participants wore an Apple Smart Watch  for 6 months (6 weeks before surgery and 24 weeks afterwards) to track their spatio-temporal parameters during early recovery via an app on an iPhone. The pre-operative metrics were statistically compared to the post-operative metrics. The authors found the variables to be poorer 2-weeks post-operatively and recover to pre-operative levels within 24 weeks. While this is an interesting and simple study, I have concerns about the accuracy and validity of the data. Please see my detailed comments below:

Conflict of interest: The role of Zimmer Biomet in this project is unclear. The company funded the study and were apparently involved in analysing the data and writing the manuscript. Further information on the involvement is required, and should perhaps be highlighted within the manuscript for transparency.

  1. This has been added to the limitations.

Introduction:

General: The background into the use of smartphones for collecting spatio-temporal parameters is lacking. Please expand this section, as this is a key focus of this study.

  1. More details regarding these parameters have been added to the introduction, lines 71-96

Line 56: One of the references cited here is irrelevant as it is a study on total hip arthroplasty

Line 58: The reference here is on total hip arthroplasty and not total knee arthroplasty

  1. The references were indeed on THA and have been removed and the manuscript revised appropriately throughout, thank you.

Line 77: It is unclear how this is a feasibility study. Please elaborate further on this point. It appears that no variables on the feasibility of using the technology or running a larger-scale study in future were measured. Thus, the aims of this study are not clear.

  1. We agree with reviewer. Our original intent was to investigate feasibility, but the analysis and paper moved to assess recovery curves.  We have removed the feasibility terminology.

Methods:

General comment: Was a sample size calculation performed for this study? If so, please include this information in the Methods. If not, please explain why.

  1. A sample size calculation was not performed, but a convenience sample was used for this secondary analysis. A sentence has been added to the methods, lines 156-157
    1. “As a secondary analysis the study used a convenience sample of all patients in the prospective study that met the inclusion and exclusion criteria”

General comment: One of my main concerns with this research study is that there is very little information on how the devices provided actually calculate the data measured. Much more detail needs to be included here so that we can understand what the authors are measuring and reporting on and whether the way in which the data was collected was reliable. It should also be clarified whether the devices have been validated in people with total knee arthroplasty, or whether all validation studies to date have been on healthy young adults.

  1. We understand why this information might be of importance to the reader. However, the biomechanical model and algorithms used to estimate walking speed, step length, double support and asymmetry percentage have not been made public by Apple Inc.
  2. We have included a brief description of each metric at lines 135-139.
  3. A validation study of Apple’s health kit data was recently published, and we have included this information in the methods section at lines 147-1 Wener et al. Sci Rep. 2023 Apr 1;13(1):5350. https://pubmed.ncbi.nlm.nih.gov/37005465/
  4. The validation study cited above contains reliability data in addition to validity data

General comment: There is almost no information on the instructions provided to patients with regards to taking part in the study. Were all participants instructed to use the devices for a certain period each day? Were they encouraged use them while exercising? What exactly were the participants told to do for the study? Without this information, it is difficult to interpret the data.

  1. Ecological validity was a main strength of this study. The data collected was minimally intrusive and representative of “real-world” behaviors. To accomplish this, patients were instructed to carry their phones near hip height and coupled with the body in a pocket or waist band whenever possible. However, no specific guidance was provided with regards frequency or duration of walking activity. Lines 138-142

Lines 56-87: It is unclear how this study relates to 'different aspects of the study'. Please explain in detail here what is meant by this statement. It is particularly important to know if the participants were asked to undertake other tasks as part of their involvement that may influence the findings of this study e.g. additional tasks related to their rehabilitation or activity.

  1. We have attempted to explain the overall premise of the study regarding these references. Additional details about the cohort study were presented in the subsequent paragraph. The 1st paragraph of the methods has been amended regarding these citations as follows: “The global study has been described in previous reports and consists of three phases including a pilot phase, a randomized controlled trial (RCT) phase, and a longitudinal cohort phase [6, 7]. These reports demonstrate data from the initial phases of the study. The current analysis utilizes patients from the cohort phase.”

Line 69 & 71: The references are irrelevant to the population in question.

  1. The reference associated with line 69 has been updated to a study done in knee OA patients, and the sentence in line 70-71 has been removed.

Line 99: What education and exercise content were patients provided with? Are these routinely administered to all patients at the study site, or were these specific to the study participants? If specific, please consider in the discussion how this may have affected the outcomes observed.

  1. Patients were to perform the therapy exercises in lieu of in person physical therapy. The exercises were standardized to each institution/study site’s standard of care, and not specific to each patient. Subjects were instructed through the app to begin therapy exercises at discharge and continue through three-months.    Lines 130-132

Lines 100-101: How was compliance reported? Was this automated i.e. through movement of the smartphone/smart watch, or did you rely on self-reported input from the participants?

  1. The metrics in this study were automatically tracked through the application. Lines 132-133

Line 104: How do the devices know if they are near the centre of mass? In other words, if the devices are not near the centre of mass, is the inaccurate data not recorded/deleted? What is the likelihood of erroneous data being included in the analyses?

  1. The iPhone and health kit are user devices, and Apple Inc. has not published their algorithms related to health kit data, thus we cannot comment on what specific conditions are required to trigger the collection of health kit data besides what has been provided by Apple (20 steps on a flat surface with the iPhone coupled to the body at hip height). We also cannot comment on erroneous data being collected beyond what has been reported by Apple in their validation study or the newly published validation study by Werner et al.
    1. Measuring Walking Quality Through iPhone Mobility Metrics. 2022, Apple Inc. p. 8-12.
    2. Werner, C., et al., Validity and reliability of the Apple Health app on iPhone for measuring gait parameters in children, adults, and seniors. Scientific Reports, 2023. 13(1): p. 5350.
  2. In order to accomplish this, patients were instructed to carry their phones close to their body (i.e.: in a pocket) at hip height.

Line 116-118: Why was recovery defined as a return to baseline when these patients were likely to have pathological walking patterns pre-operatively?

  1. We agree that improvement beyond baseline is the ultimate goal of TKA. However, in the immediate post-operative period, TKA resulted in reduced capabilities, as evidenced by the significant reduction in gait metrics at week two in our data and consistent with published reports on the recovery curve[1]. Therefore, we defined recovery from the surgery as a return to pre-operative (baseline) capabilities. Further, prior studies assessing step count recovery curves following TKA have referred to recovery as the time point when post-operative step counts were similar to pre-operative step counts [2-4]. Therefore, we believe there is precedent to operationally define recovery as recovery from the procedure. Given the precedence, we do not feel it necessary to revise our use of the term recovery.

Results:

Figure 1: It is unclear to me why patients who did not have a Zimmer-Biomet implant were excluded from this analysis, as the research question appears to be completely unrelated to the implant used. This raises conflict of interest issues and ethical issues with regards to participant inclusion in the study.  Please clarify further.

  1. The initial inclusion criteria of the study were limited to Zimmer Biomet implants, prosthesis type. In a subsequent amendment the application was made device agnostic and as such the study allowed additional devices. However, those devices were added near the end of the enrollment period for this secondary analysis and as such many were excluded for not having the minimum 24-week follow-up.We have re-run the analysis with all patients meeting the other inclusion/exclusion criteria. The results have been updated and the paper is now device agnostic. The patient attrition flow chart has been simplified to exclude these cases in the exclusion criteria for <24-week follow-up.
  2. The inclusion of non-Zimmer Biomet implants has increased the sample size by 210 cases to 686 cases, but had negligible effects on the results and recovery curves.

Discussion:

Line 158: The reference is not relevant to the population in question.

  1. The reference has been updated to Baczkowicz et al. Gait and functional status analysis before and after total knee arthroplasty [2]

Conclusion:

General comment:  It is difficult to know the validity of the conclusion, as the methods by which the data were collected are not described clearly enough in the manuscript.

  1. We hope that by addressing the methodological concerns the reviewer can better interpret the validity and appropriateness of the conclusion statements.

A

References

  1. Anderson, M., et al., THE RECOVERY CURVE FOR PHYSICAL ACTIVITY FOLLOWING PRIMARY KNEE ARTHROPLASTY USING PASSIVELY COLLECTED OBJECTIVE MEASURES WITH A SMARTPHONE-BASED CARE PLATFORM AND SMART WATCH. Orthopaedic Proceedings, 2021. 103-B(SUPP_9): p. 15-15.
  2. BÄ…czkowicz, D., et al., Gait and functional status analysis before and after total knee arthroplasty. Knee, 2018. 25(5): p. 888-896.
  3. Lebleu, J., et al., Predicting physical activity recovery after hip and knee arthroplasty? A longitudinal cohort study. Braz J Phys Ther, 2021. 25(1): p. 30-39.
  4. Twiggs, J., et al., Measurement of physical activity in the pre- and early post-operative period after total knee arthroplasty for Osteoarthritis using a Fitbit Flex device. Med Eng Phys, 2018. 51: p. 31-40.
  5. Constantinou, M., et al., Spatial-temporal gait characteristics in individuals with hip osteoarthritis: a systematic literature review and meta-analysis. J Orthop Sports Phys Ther, 2014. 44(4): p. 291-b7.
  6. Crawford, D.A., et al., 2021 Mark Coventry Award: Use of a smartphone-based care platform after primary partial and total knee arthroplasty: a prospective randomized controlled trial. Bone Joint J, 2021. 103-B(6 Supple A): p. 3-12.
  7. Tripuraneni, K.R., et al., A Smartwatch Paired With A Mobile Application Provides Postoperative Self-Directed Rehabilitation Without Compromising Total Knee Arthroplasty Outcomes: A Randomized Controlled Trial. J Arthroplasty, 2021. 36(12): p. 3888-3893.

Reviewer 4 Report

more keywords should be included to reach more readers

the flowchart needs to be modified, it is not of good quality

It would be necessary to improve the quality of English so that readers can better understand some words.

Author Response

Reviewer 3: We have reviewed your comments and hope that our responses are sufficient to resolve your concerns.

more keywords should be included to reach more readers

  1. We have added the following: sensors, recovery curve, accelerometers

the flowchart needs to be modified, it is not of good quality

  1. We have inserted another figure, though the quality is only marginally better due to copy and pasting into Microsoft Word. The editorial office has been provided with a file of high-quality images for each figure at submission, and an updated copy with the revised manuscript.

It would be necessary to improve the quality of English so that readers can better understand some words.

  1. The manuscript has now been proofread by 8 native English speakers.

Reviewer 5 Report

Thank you for the opportunity to review this interesting manuscript.

Before of all, please justify the right margen.

Line 19. Please change "speeds" by "speed".

I consider that p-value must be written as "p=0.xxx".

Authors use "total knee arthropasty" or "total joint arthrosplasty". If they refer to the same, please, use the same term always.

Please, rewrite the objective. In line 79 you write "total knee arthroplasty" and in line 80 "TKA". Please, be consistent with the use of abbreviations.

In methods, line 116, explain the variables.

The quality of figure 1 is low. Please, increase it. 

Caption of the figure should be placed under the figure.

I think that "cadence" is a variable that needed to be assessed.

I consider that English language edition would be recommendable.

Author Response

Reviewer 4: Your review is much appreciated, and your thoughtful comments have been thoroughly discussed. The following responses were felt appropriate, and we hope you are able to come to the same conclusion. Thank you for your time in helping to improve this manuscript.

Thank you for the opportunity to review this interesting manuscript.

Before of all, please justify the right margen.

  1. The margin was set as part of the Journal’s formatting and upon their request.

Line 19. Please change "speeds" by "speed".

  1. Done, thank you.

I consider that p-value must be written as "p=0.xxx".

  1. Edited, thank you

Authors use "total knee arthroplasty" or "total joint arthroplasty". If they refer to the same, please, use the same term always.

  1. The paper has been reviewed in detail for this and corrections made when appropriate.

Please, rewrite the objective. In line 79 you write "total knee arthroplasty" and in line 80 "TKA". Please, be consistent with the use of abbreviations.

  1. The paper has been reviewed in detail for this and corrections made where appropriate.

In methods, line 116, explain the variables.

  1. An addition has been added to line 132-139 to indicate the outcome variables

The quality of figure 1 is low. Please, increase it. 

  1. We have inserted another figure, though the quality is only marginally better due to copy and pasting into Microsoft Word. The editorial office has been provided with a file of high-quality images for each figure at submission and again with the revision.

Caption of the figure should be placed under the figure.

  1. Edited

I think that "cadence" is a variable that needed to be assessed.

While cadence would have been an interesting variable, it was not included in our data collection and thus we cannot assess it. Cadence is often used as an indirect proxy for exercise intensity, as is walking speed. Given that walking speed is a product of cadence and step length, and because walking speed recovered but step length did not, a reader interested in cadence may interpret this information as an increase in cadence at 24 weeks post-operative. 

Round 2

Reviewer 2 Report

Thank you for making the changes within the manuscript and for taking the time to respond to my comments in the cover letter. 

I have no further remarks on this manuscript and I advise to accept the paper for publication in present form.